# EPFL-Smart-Kitchen: An Ego-Exo Multi-Modal Dataset for Challenging Action and Motion Understanding in Video-Language Models

**Andy Bonnetto[1,*], Haozhe Qi[1,*], Franklin Leong[1], Matea Tashkovska[1],**
**Mahdi Rad[2], Solaiman Shokur[1,3], Friedhelm Hummel[1,4,5,6],**
**Silvestro Micera[1,3], Marc Pollefeys[2,7], Alexander Mathis[1,✉,*]**

1: École Polytechnique Fédérale de Lausanne (EPFL), Lausanne
2: Microsoft 3: Scuola Superiore Sant'Anna, Pisa
4: Swiss Federal Institute of Technology Valais (EPFL Valais), Sion
5: Clinique Romande de Réadaptation, Sion
6: University of Geneva Medical School, Geneva
7: Eidgenössische Technische Hochschule (ETH), Zürich

## Abstract

Understanding behavior requires datasets that capture humans while carrying out complex tasks. The kitchen is an excellent environment for assessing human motor and cognitive function, as many complex actions are naturally exhibited in kitchens from chopping to cleaning. Here, we introduce the EPFL-Smart-Kitchen-30 dataset, collected in a noninvasive motion capture platform inside a kitchen environment. Nine static RGB-D cameras, inertial measurement units (IMUs) and one head-mounted HoloLens 2 headset were used to capture 3D hand, body, and eye movements. The EPFL-Smart-Kitchen-30 dataset is a multi-view action dataset with synchronized exocentric, egocentric, depth, IMUs, eye gaze, body and hand kinematics spanning 29.7 hours of 16 subjects cooking four different recipes. Action sequences were densely annotated with 33.78 action segments per minute. Leveraging this multi-modal dataset, we propose four benchmarks to advance behavior understanding and modeling through 1) a vision-language benchmark, 2) a semantic text-to-motion generation benchmark, 3) a multi-modal action recognition benchmark, 4) a pose-based action segmentation benchmark. We expect the EPFL-Smart-Kitchen-30 dataset to pave the way for better methods as well as insights to understand the nature of ecologically-valid human behavior. Code and data are available at https://amathislab.github.io/EPFL-Smart-Kitchen.

## 1 Introduction

Understanding human behavior is fundamental across multiple domains - from augmented reality[9] and robotics [18] to neuroscience [52, 54] and neuroengineering [56]. While we have made significant progress in behavioral analysis through action recognition [89, 37, 83, 17], action segmentation [86, 45, 98, 77] and motion generation [80, 99, 27], critical gaps remain. Current datasets face a fragmentation problem (Table 1). Existing datasets excel in isolated aspects of behavioral capture, but lack integration. Some datasets advance full-body 3D pose estimation but provide insufficient hand tracking for complex movements. Others offer detailed finger articulations but are limited to constrained environments, missing the crucial full-body context including the global

---

*A.B. and H.Q. contributed equally. Correspondence: `alexander.mathis@epfl.ch`

39th Conference on Neural Information Processing Systems (NeurIPS 2025) Track on Datasets and Benchmarks.

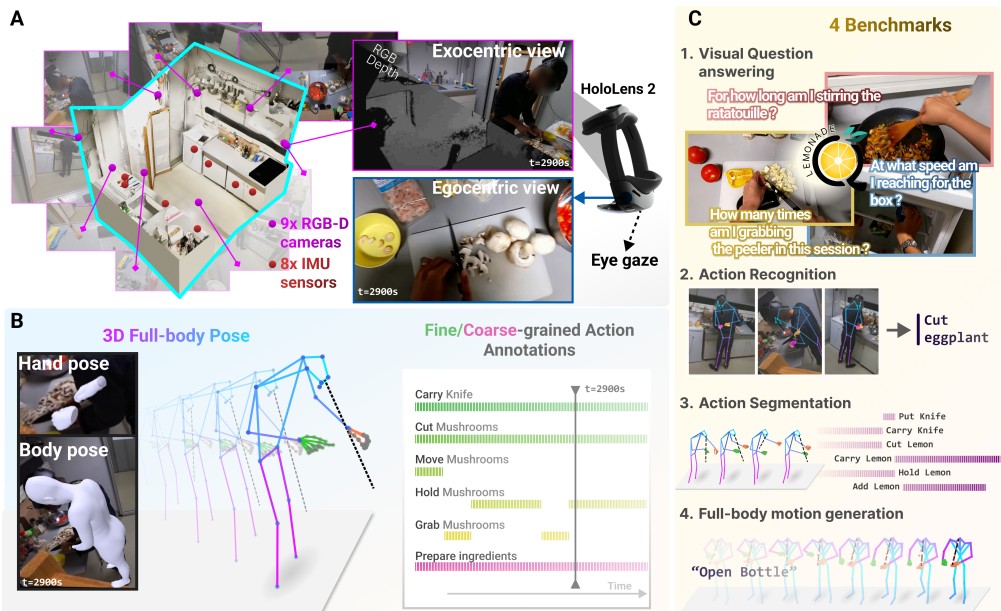

Figure 1: **The EPFL-Smart-Kitchen-30, dataset and benchmarks.** (A) **Collected data**. 3D kitchen reconstruction, purple points are fixed RGB-D cameras. Subjects cook with a HoloLens 2 headset recording both egocentric videos and eye gaze. (B) **Extracted data**. 3D body and hand poses are extracted from multiple data sources. Fine-grained and coarse-grained action segments are densely annotated. (C) **Benchmarks.** We propose four benchmarks based on the EPFL-Smart-Kitchen-30 dataset. A Visual question answering benchmark, an action segmentation benchmark, an action recognition benchmark and a full-body motion generation benchmark.

position. Moreover, some datasets fail to capture two essential components of natural behavior: goal-directed actions and eye movements. Human actions are inherently purposeful and guided by visual attention [28], yet most current datasets do not incorporate these elements, resulting in an incomplete representation of behavior. Comprehensive datasets of full-body, including hand and eye tracking alongside synchronized multi-view video and detailed, hierarchical action annotations are currently missing, and significantly hinder our ability to analyze natural human behavior [28, 52, 77].

We present the EPFL-Smart-Kitchen-30, a dataset that captures humans in authentic cooking scenarios with multimodality. It features both egocentric and exocentric perspectives through ten synchronized camera views, providing excellent visual coverage of natural cooking behaviors. EPFL-Smart-Kitchen-30 includes multiple modalities: RGB and depth images, IMU data, eye gazes, and 3D hand/body poses (Figure 1A-B).The EPFL-Smart-Kitchen-30 compares favorably to other datasets (Table 1) and promises to advance multimodal fine-grained action understanding. The scale is substantial: 29.7 hours of multi-view, multimodal recordings from 16 participants across 49 complete cooking sessions, from recipe reading to cleanup. The dataset defines 763 fine-grained actions, ensuring dense, hierarchical action annotation and exclusive action definitions. Sessions are densely annotated, yielding 55,361 fine-grained action segments and 4,828 coarse-grained activity segments—about 33 actions per minute.

With the annotated data, we build four benchmarks for action understanding and modeling (Figure 1C). First, we introduce Lemonade, a novel approach that transforms our ground truth annotations and pose estimations into challenging close-ended question-answer pairs (QA). This benchmark specifically tests the behavioral understanding capabilities of video-language models (VLMs). Second and third, we propose action recognition and segmentation benchmarks that span multiple modalities, providing empirical insights into procedural human behavior. Fourth, we present a full-body motion generation benchmark that highlights EPFL-Smart-Kitchen-30's unique value for generative tasks, demonstrating how our integrated data approach enables more natural and contextually appropriate motion synthesis. In summary, we make the following contributions (Figure 1):

- We capture 30 hours of goal-directed cooking behavior from ego-exo perspectives

- We densely annotate fine-grained actions and coarse-grained activities.
- We propose multimodal behavior understanding (action recognition, segmentation and vision-language question answering) and modeling benchmarks

These contributions collectively address the fragmentation problem in behavioral analysis and provide the research community with new possibilities for integrated, context-rich human behavior understanding.

## 2 Related work

### 2.1 Datasets of human behavior

Many datasets have been proposed that record participants executing purposeful motions [72, 47, 65], which were further extended to fitness activities by datasets like EgoExo-Fitness [41] and FLAG3D [78] to include more complex human body motions. Traditional motion capture approaches focused on isolated movements, offering high controllability but sacrificing critical contextual information. Recent research has shifted toward recording behavior in natural settings, enabling the study of authentic transitions and sequence patterns that characterize genuine human activity. In particular, absolute positioning determines the agent's spatial location and facilitates the analysis of its interactions with the environment [49]. Assembly-based datasets collect structured object interactions [6, 91, 3, 70], but by using a greater variety of actions and natural environments, cooking is getting popular for building action datasets such as EPIC-KITCHENS-100 [14], Humans in kitchens [79] or certain sequences of the large EgoExo4D [24] dataset. The EPFL-Smart-kitchen-30's dense annotations suit the characterization of body and hand movement transitions and distinguish themselves with their unambiguous and rich action descriptions (Table 1).

Language can flexibly describe behavior, and VLMs promise to capture that richness. The general video understanding of VLMs has been evaluated with exhaustive benchmarks such as MVBench [38] and Video-MME [20]. More specific challenges subsequently developed to tackle long-term understanding [105, 51] and egocentric video understanding [51, 30]. In the case of behavior understanding, ActivityNet-QA [96] and NExT-QA [94] evaluate the causal and temporal abilities of VLMs. While subsets of certain benchmarks [13, 38, 105] contain questions related to motion, they mostly focus on understanding behavior at the event level. EPFL-Smart-Kitchen-30 enables a fundamentally different approach. Our Lemonade benchmark introduces questions that specifically probe the understanding of human kinematics and fine-grained behavioral details that previous datasets simply cannot address.

Table 1: **Action dataset comparison.** # indicates "number of" for simplicity. The remaining columns mark following features: parametric model for motion representation (PM), structured actions (SA), markerless video recording (ML), depth recording (DR), and absolute positioning (AP). AP refers to global positioning (derived from point clouds) that is consistent across different sessions. Note: EgoExo4D [24] reports 1422h by summing per camera recording time, where the total activity is 180h, yet only 88.8h annotated with MSCOCO keypoints (numbers from [49]).

| | Datasets | Total hours | Duration (min) | # segments | Seg. per min | # action classes | # Ego/ Exo | Body PM | Hand PM | Eye Gaze | SA | ML | DR | AP |
|---|---|---|---|---|---|---|---|---|---|---|---|---|---|---|
| Video-focused | Meccano [67] | 6.9 | 20.7 | 8,857 | 21.4 | 61 | 0/1 | ✗ | ✗ | ✓ | ✓ | ✓ | ✓ | ✗ |
| | IKEAASM [6] | 11.7 | 1.9 | 17,577 | 8.4 | 33 | 0/3 | ✗ | ✗ | ✗ | ✓ | ✓ | ✓ | ✓ |
| | EPIC-100 [14] | 100.0 | 8.6 | 89,977 | 15.0 | 4,053 | 1/0 | ✗ | ✗ | ✗ | ✓ | ✓ | ✗ | ✗ |
| | EgoExo4D [24] | 180 | 15.3 | 20,406 | 4.5 | 689 | 1/4-5 | ✗ | ✗ | ✓ | ✓ | ✓ | ✗ | ✗ |
| | HoloAssist [91] | 166.0 | 4.5 | 184,838 | 18.6 | 1,887 | 1/0 | ✗ | ✗ | ✓ | ✓ | ✓ | ✓ | ✓ |
| | EgoExo-Fitness [41] | 32 | 1.5 | 6,131 | 4 | 12 | 0/4 | ✗ | ✗ | ✗ | ✓ | ✓ | ✗ | ✗ |
| Motion-focused | AMASS [50] | 43 | 0.22 | 11,451 | 0.22 | - | - | ✓ | ✓ | ✗ | ✗ | ✗ | ✗ | ✗ |
| | BABEL [65] | 43 | 0.39 | 28,000 | 10.7 | 250 | - | ✓ | ✗ | ✗ | ✓ | ✗ | ✗ | ✗ |
| | HumanML3D [26] | 28.6 | 0.12 | 14,616 | 8.5 | - | - | ✓ | ✗ | ✗ | ✗ | ✗ | ✗ | ✗ |
| | HumanAct12 [25] | - | - | 1,191 | - | 34 | - | ✓ | ✗ | ✗ | ✓ | ✓ | ✓ | ✗ |
| Video-motion focused | Assembly101 [70] | 41.8 | 7.1 | 84,460 | 33.1 | 1,380 | 4/8 | ✗ | ✗ | ✗ | ✓ | ✓ | ✗ | ✓ |
| | H2O [34] | 5.5 | 0.33 | 1,000 | 3.0 | 36 | 1/4 | ✗ | ✓ | ✗ | ✓ | ✓ | ✓ | ✗ |
| | MotionX [43] | 144 | 0.11 | 81,100 | 9.4 | - | 0/1 | ✓ | ✓ | ✗ | ✗ | ✗ | ✗ | ✗ |
| | Nymeria [49] | 300 | 15 | - | - | - | 1/1 | ✓ | ✗ | ✓ | ✗ | ✗ | ✗ | ✓ |
| | **EPFL-Smart-Kitchen-30** | 29.7 | 35.9 | 60,189 | 33.78 | 768 | 1/9 | ✓ | ✓ | ✓ | ✓ | ✓ | ✓ | ✓ |

By leveraging our multimodal data integration, we can evaluate models on their ability to reason about body movements and hand-object interactions (Figure 4).

## 2.2 Models for behavior understanding

The ability to predict movement patterns provides a valuable approach to understanding behavior. Movement can be captured in various forms, including video recordings, pose estimation data, and IMU recordings. Improvements in deep learning models together with their increase in computational power levels have led to the development of many multi-view, multimodal action understanding algorithms [71, 88, 104, 7, 93, 73]. Shah et al. [71] leverage contrastive learning to align the feature spaces from different views. Wang et al. [88] use an adversarial generative network to constrain RGB and depth modality information. HandFormer [73] combines 3D hand poses and RGB frames together for action recognition. LaViLa [104] learns video representations from pre-trained large language models. TIM [7] designs time interval encodings to incorporate visual and audio events. Despite progress, current methods are limited in views and modalities, partially due to the lack of large-scale multi-view, multimodal action datasets. With our EPFL-Smart-Kitchen-30 dataset, we set up multi-view, multimodal action understanding benchmarks taking and comparing exocentric videos, egocentric videos, full-body pose estimations, and eye gaze modalities as input, with the possibility to also include depth videos and IMU recordings.

Another approach for behavior understanding is through the ability to generate movement of a target behavior. Recently, text-to-motion generation gained a lot of attention [80, 8, 99, 27, 81, 63, 101]. We propose a novel semantic text-to-motion generation benchmark that considers full-body pose representations, including eye gaze. for situated motion generation. This contrasts with the commonly used KIT [64] and HumanML3D [26], which do not incorporate hand models or gaze information.

By integrating language, VLMs provide more flexible ways to understand behavior. VideoL-LaMA3 [97] captures fine-grained details and temporal dynamics in videos through its dynamic resolution mechanisms and advanced positional embedding strategies, whereas Qwen2.5-VL [5] and Intern VL2.5 [11] better integrate multimodal inputs. Specific tasks such as long-term video understanding usually rely on video compression [36, 74, 40, 42] or on extending their context length [102, 92, 10, 46]. We challenge these models to operate beyond their conventional performance by proposing a benchmark that leverages behavioral context and kinematics.

## 3 The EPFL-Smart-Kitchen-30

Here we introduce the EPFL-Smart-Kitchen-30 dataset, which features multi-view, multimodal data of human cooking with fine-grained and coarse-grained action annotations (Figure 1B). We will

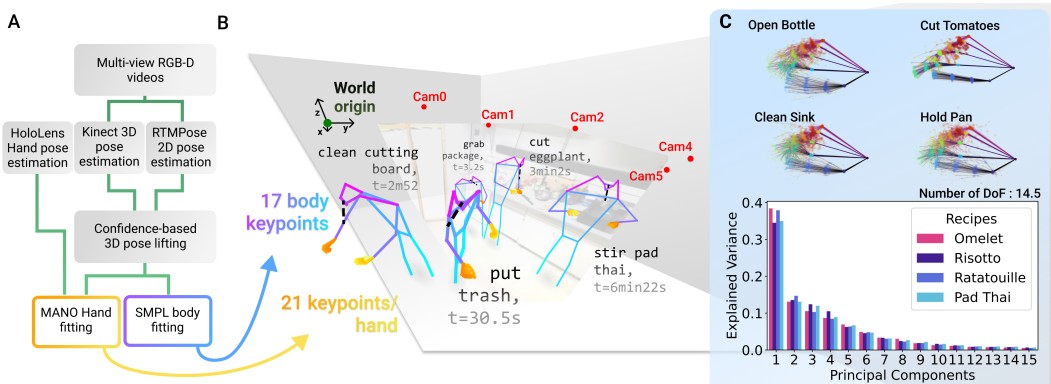

Figure 2: **Full-body 3D pose estimation** (A) Pipeline for 3D pose estimation (B) Poses and camera positions are defined relative to a global coordinate system comparable in the same environment. We illustrate several reprojected 3D poses on camera 6. (C) Characterization of right-hand poses to show the captured kinematic diversity: (Top) examples of hand poses for four actions, (Bottom) Number of principal components necessary to capture the right-hand poses during cooking for the Exo-Hand in each recipe, around 14 degrees of freedom (DoF) are necessary to explain 95% of the variance.

describe the setup (Sec. 3.1) and the data collection procedure (Sec. 3.2). Then, we illustrate the 3D pose regression (Sec. 3.3) and detail the action annotation characteristics (Sec. 3.4).

## 3.1 The EPFL-Smart-Kitchen setup

Capturing multi-view, multimodal data is a challenging task that requires the synchronization and calibration of multiple sensors. To capture naturalistic cooking behaviors, we built the EPFL-Smart-Kitchen, a fully functional kitchen with appliances and utensils. Cooking materials, including pots, pans, and other utensils, were provided to the subjects along with the ingredients and spices necessary for preparing the recipe (Supp. Sec. A).

To minimally affect the subjects' natural movements while capturing multimodal information, we installed nine Microsoft Kinect Azure RGB-D cameras [57] at strategic points inside the kitchen, four focusing on the global exocentric view and five focusing on local exocentric views (counters, stove, and sink, see Figure 1A). We additionally equipped the kitchen with eight IMU sensors on the frequently used equipment (e.g., fridge door, five cupboard doors, knife, and spatula). Subjects wore a Hololens 2 headset [85], a mixed-reality headset that can capture egocentric views and eye gaze data under global calibration. We synchronized all devices using audio signals and a trigger, and calibrated all the cameras (Supp. Sec. A.2).

## 3.2 Data collection procedure

To capture realistic cooking scenarios, subjects prepared a meal from reading a recipe to cleaning up, leading to significantly longer recordings than in most existing action datasets (Table 1). We recruited 16 subjects (four males and twelve females, two left-handed, ages 20-46) to cook for up to five sessions in the EPFL-Smart-Kitchen (Supp. Sec.A.3). During each session, subjects are asked to follow one of four different recipes (omelet, pad thai, risotto, ratatouille), adapted to their preferences and requirements. Overall, we recorded and processed 29.7h of cooking experiments, corresponding to 3,207,600 frames per camera for 49 cooking sessions. All procedures were approved by the EPFL-Ethical Board. Subjects' faces are anonymized across all videos to address privacy concerns. All subjects consented and were informed about the ethics (Supp. Sec. D.1).

## 3.3 Estimation of 3D motions

We placed the cameras so that both the body and the hands of the participants are visible from at least three angles. Four cameras captured global body information, while five cameras captured local hand information. Using multi-view RGB-D video, we conduct body/hand mesh fitting and tracking using all 10 camera views, extracting 2D and 3D pose information from each view with existing pose estimation tools. Specifically, we extract 2D body and hand poses using RTMPose [31], available in DeepLabCut v3 [53], 3D body poses and tracklets using the Kinect body tracking SDK [58], and 3D hand poses using the HoloLens 2 hand tracking toolkit [59]. We lift 2D poses to 3D poses and fit the SMPL [62] body mesh by minimizing the 3D joint, 2D reprojection, temporal smoothing, and regularization loss, as well as the hand 3D joint loss to fit the MANO hand mesh [68] (Figure 2A and Suppl. Sec. C). The average absolute error compared to triangulated manually-annotated 2D poses is $6.22cm \pm 5.16cm$ and $3.30cm \pm 5.12cm$ for the body and hand respectively, our margins are comparable with those of [24, 34] (Supp. Sec. C.5).

To illustrate the richness of the captured movement data (Figure 2B-C), we estimate the number of degrees of freedom (kinematic synergies) in pose space based on a common method in neuroscience [82, 69, 12]. We found that the dataset exhibits a large number of degrees of freedom (Figure 2C), which foreshadows the potential for studies on human behavior.

## 3.4 Annotation of fine-grained actions and coarse activities

The annotated action classes were defined with the following considerations. Firstly, many contemporary datasets (e.g.,[14, 91]) tend to allow the annotators to freely describe the actions and then post-hoc group the actions based on action similarity. This might lead to different names for similar actions (e.g., *pour* and *fill* [14]) and thus introduce ambiguity. We instead curated a set of verbs and nouns. We annotated with temporal overlaps between actions. For example, when labeling *cut tomato*, we also label *carry knife* and may label *hold tomato*. This enriches the annotation at a given

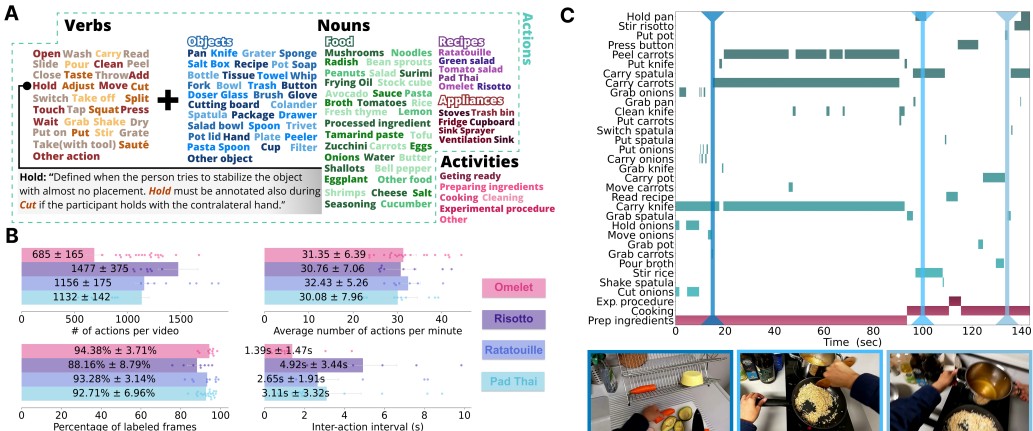

Figure 3: **Hierarchical action annotations**: (A) List of verbs, nouns and activities used for action annotation, each action (verbs) is specifically defined as shown for "Hold". (B) Statistics of annotation segments for fine-grained actions. (C) Example ethogram with selected egocentric frame to illustrate the richness of the actions (turquoise) and activity (pink) annotations comprising short and long segments that can overlap.

timestep and attempts to reduce ambiguity. Based on the above rationale, we define 33 verbs and 79 nouns, which compose 763 fine-grained actions. Each verb is defined by a rule-based description intended to prevent confusion (see example in Figure 3A and Supp. Sec. B). During the annotation procedure, we asked annotators to watch videos and annotate the start and stop times of actions. To define the behavioral contexts for each action, six coarse-grained exclusive activities were annotated, summing up to 4,828 segments. Thus, behavior is annotated in a hierarchical fashion [4, 77, 21].

The quality and reliability of annotations were validated following the protocol outlined in Supp. Sec. B.5 In total, 60,189 action segments were annotated, resulting in 33.78 action segments per minute (Figure 3B). The richness of the action annotation is demonstrated by a large variety of action lengths (from 1 second to 100 seconds, Figure 3C). Overall, the different views and modalities contribute to fine-grained action understanding in different aspects: 1) RGB frames focus on coarse-grained information while depth frames rather focus on geometric aspects; 2) the egocentric view and the eye gaze data captured from the HoloLens 2 are related to (part of) what the subject sees; 3) global exocentric views and the body poses capture the overall context; 4) local exocentric views, hand poses, and tool IMU data capture fine-grained movements and hand-object interactions.

## 4 Multimodal action and motion understanding benchmarks

Cooking involves many different actions that are sequenced in a goal-directed fashion to achieve a tasty outcome. In each experimental session, subjects go from reading the recipe and preparing the ingredients to creating the dish and ultimately cleaning up. To make progress towards analyzing such complex human behavior, we created four behavior understanding benchmarks (Figure 1C). Two benchmarks focus on multimodal behavior analysis: action recognition and action segmentation. These benchmarks are complemented by a full-body behavior synthesis benchmark (motion generation). Furthermore, we designed a question-answering benchmark (Lemonade) to understand human cooking behavior. Lemonade is structured for zero-shot evaluation. For the other benchmarks, we split the sessions into train, validation, and test sets. The training/validation sets are split into 26/7 sessions chosen so that every recipe is present in the validation set and to balance the number of rare action segments in both sets. The test set is composed of 16 sessions which also include new subjects. The curated dataset used for the benchmark excludes actions with less than 3 instances and is composed of 31 verbs, 78 nouns, and 581 actions together with six activities.

### 4.1 Lemonade: Language models Evaluation of MOtion aNd Action-Driven Enquiries

**Rationale.** VLMs exhibit remarkable potential for understanding human behavior [90, 100, 36, 95]. They raise intriguing questions: Can they accurately predict preceding or subsequent actions in behavioral sequences? Are they able to infer long-term behavioral patterns from just a few frames?

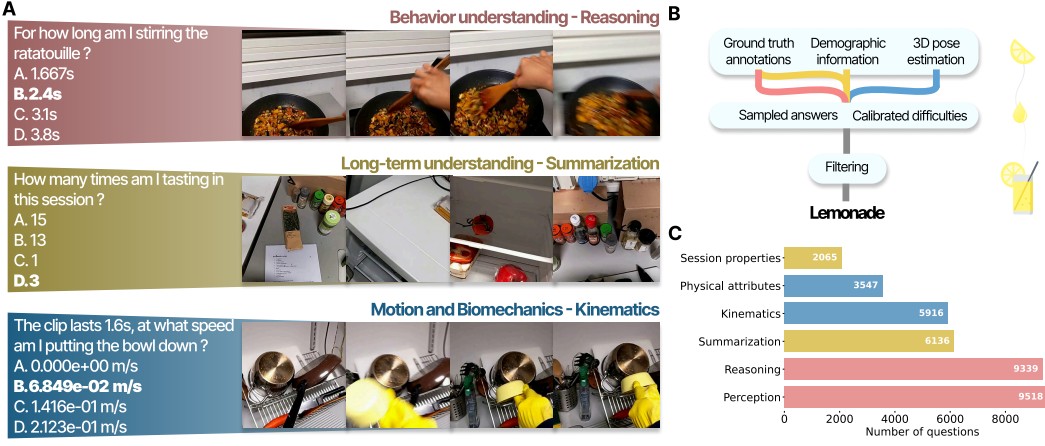

Figure 4: **Lemonade**: (A) Examples of video question pairs for each category. More examples in Supp. Sec. F.1 (B) Questions are designed from ground truth annotations. (C) Distribution of questions for all subcategories.

Furthermore, can their general knowledge enable precise distance and velocity estimations from video data alone? EPFL-Smart-Kitchen-30 provides the ideal testbed to explore these fundamental questions about machine understanding of natural human behavior (Figure 4A). Thus, we introduce Lemonade: **L**anguage models **E**valuation of **MO**tion a**N**d **A**ction-**D**riven **E**nquiries. The Lemonade framework (Figure 4B) is designed to generate millions of unique QA pairs by combining various video clips, question formats, and answer types. Lemonade consists of 36,521 closed-ended QA pairs linked to egocentric video clips, categorized into three groups and six subcategories (Figure 4C). 18,857 QAs focus on behavior understanding, leveraging the rich ground truth behavior annotations of the EPFL-Smart-Kitchen to interrogate models about perceived actions (Perception) and reason about unseen behaviors (Reasoning). 8,201 QAs involve longer video clips, challenging models in summarization (Summarization) and session-level inference (Session properties). The remaining 9,463 QAs leverage the 3D pose estimation data to infer hand shapes, joint angles (Physical attributes), or trajectory velocities (Kinematics) from visual information. More examples and details on QA design can be found in the Supp. Sec. F.1

**Baselines and Metrics.** Based on state-of-the-art results from other recent benchmarks [84], we evaluated three open-source and one closed-source VLM SoTA models, namely InternVL2.5 [11], LLaVA-OneVision [36], Qwen2.5-VL [5] and Gemini 2.0 Flash [15]. To ensure consistent evaluation and enable future comparisons, all models are evaluated using lmms-eval [100], where Lemonade is implemented as a new evaluation task. To interpret the results, we manually answered 1,662 questions. As is commonly done for question answering, we evaluate the model performance as average accuracy [20, 38, 94, 96].

**Results.** Different VLMs achieved high accuracy in identifying ongoing actions and activities, as well as predicting immediate next and previous actions. However, Lemonade exposed critical limitations in VLMs in predicting general context information, distances, timings, and body kinematics (Table 2). However, this benchmark is also challenging for humans. Merely relying on visual input and language proved insufficient for these precise kinematic estimations, which require accurate frame timing and depth reference data. To overcome these challenges, future models could benefit from explicitly integrating additional modalities, which is becoming possible with multimodal language models [48, 19].

## 4.2 Action recognition benchmark

**Rationale**. Given a trimmed action segment, the action recognition model needs to predict the corresponding action class [89, 37, 83, 17]. We built a fine-grained action recognition benchmark for 763 classes with a long-tailed distribution and allow different data types as input. Specifically, we formulated flexible masked auto-encoding baselines taking the egocentric view, one exocentric view, the 3D body poses, the 3D hand poses, and the eye gaze rays as input and compared different combinations of those data sources.

Table 2: Accuracy of VLMs (8 frames) on the lemonade benchmark per category; chance level is 25%. Human carried out answered 1,662 samples. Detailed results in Supp. Table. F.2

| Models | Perc. | Reas. | Sess. | Summ. | Phys. | Kin. | All |
|---|---|---|---|---|---|---|---|
| Human* | 62.32 | 52.38 | 62.38 | 44.40 | 70.78 | 38.34 | 54.75 |
| Gemini-2.0-Flash | 41.47 | 41.03 | **49.10** | 35.09 | **40.26** | 21.69 | 37.39 |
| LLaVa-OneVision | **47.23** | 41.46 | 36.76 | 41.87 | 34.03 | **34.09** | **40.85** |
| InternVL2 5-8B | 43.94 | 41.52 | 45.76 | 33.80 | 39.89 | 27.74 | 38.70 |
| Qwen2.5-VL-32B (4fr.) | 43.72 | **45.87** | 46.25 | **42.75** | 32.22 | 28.52 | 40.67 |
| Qwen2.5-VL-7B | 42.86 | 43.58 | 48.14 | 37.66 | 38.93 | 25.64 | 39.30 |

**Baselines and Metrics.** We adapted VideoMAE [83, 21] model into a multi-modal MAE, enabling it to take multi-view videos, 3D poses, and eye gazes as inputs (Supp. Sec. F.4). Like others, we evaluate the model performance by Top-1 and Top-5 accuracy for verb, object, and action classes [91, 70]. Considering that the number of action samples across different actions has a long-tail distribution, we selected the top 180 most frequent actions as head actions and report the performance for both head actions and tail actions separately.

**Results**. Simple concatenation of all modalities as inputs yielded slightly better performance compared to single video modality (Table 3). Meanwhile, transfer learning from a ViT model trained on EPIC-KITCHENS-100 [14] boosted the performance for baselines with visual inputs and reduced the performance for pose only. Furthermore, adding (hand and body) pose information to the video-based models yielded better overall performance, which mainly benefits from the verb prediction improvements. However, simply concatenating tokens from different modalities barely improved the performance. To efficiently utilize multiple modalities without significant computational cost, we integrate egocentric view, multi-exocentric views, and hand pose data (⚙) (Supp. Sec. F.4.4) to boost the performance by 21.6% when trained from scratch (Supp. Table F.3) and 6.3% when trained from the pretrained model, over the egocentric-only model. Overall, we hope this benchmark will inspire the community to create models that can effectively use multiple modalities.

Table 3: **Fine-grained action recognition benchmark results from pretrained model.** 📷: egocentric view, ▶: global exocentric view, 🧍: 3D body pose, ✋: 3D hand pose, 👁: eye gaze, ⚙(✋×multiple ▶) : hand cropped videos. Combining modalities has the potential to increase the performance. Our best results are achieved by cleverly merging these modalities together.

| Modalities | All Classes Accuracy Top1/5 | | | Head Classes Accuracy Top1/5 | | | Tail Classes Accuracy Top1/5 | | |
|---|---|---|---|---|---|---|---|---|---|
| | Action | Verb | Noun | Action | Verb | Noun | Action | Verb | Noun |
| 📷 | 37.51/62.94 | 57.72/92.18 | 52.03/79.05 | 41.12/67.00 | 59.74/93.36 | 55.62/82.11 | 16.64/39.51 | 46.06/85.35 | 31.27/61.38 |
| 📷🧍 | 37.87/63.56 | 58.90/93.59 | 52.56/79.64 | 41.55/67.67 | 60.56/94.66 | 56.43/82.84 | 16.66/39.85 | 49.28/87.44 | 30.19/61.11 |
| 📷▶ | 37.57/63.13 | 58.38/92.92 | 52.29/78.45 | 41.14/67.31 | 60.43/93.90 | 55.90/81.62 | 16.97/39.00 | 46.54/87.24 | 31.46/60.15 |
| 🧍✋ | 11.80/25.49 | 38.67/78.80 | 19.83/42.23 | 13.55/28.77 | 39.87/80.65 | 22.31/46.29 | 1.70/6.57 | 31.77/68.15 | 5.49/18.79 |
| 📷🧍✋ | 38.31/64.75 | 60.41/93.78 | 52.51/79.91 | 41.83/68.89 | 61.96/94.79 | 56.27/83.14 | 17.97/40.90 | 51.45/87.94 | 30.81/61.27 |
| 📷🧍✋👁 | 37.35/62.34 | 60.78/93.04 | 50.58/77.30 | 40.91/66.05 | 62.20/93.92 | 54.15/80.48 | 16.85/40.97 | **52.55**/87.94 | 30.02/58.99 |
| 📷▶🧍✋👁 | 37.49/62.76 | **61.04**/93.59 | 50.94/77.52 | 41.09/66.50 | 62.53/94.51 | 54.75/80.57 | 16.66/41.21 | 52.42/**88.29** | 28.95/59.91 |
| 📷⚙(✋× ▶) | **40.03/67.01** | 60.80/**94.65** | **55.38/82.58** | **43.60/71.25** | 62.60/95.76 | 59.00/85.62 | **19.44/42.52** | 50.41/88.25 | **34.48/65.02** |

### 4.3 Action segmentation benchmark

**Rationale**. Given an untrimmed video, action segmentation requires the model to predict one or multiple action classes for every frame [86, 45, 98, 77]. Given the absence of popular (and comprehensive) action segmentation benchmarks from 3D pose data (Section 2), we built an action segmentation benchmark that compares the impact of different input data (body, hand, eyes, video features). One might expect that actions such as moving through the kitchen will be better predicted from the body, while motions like cutting require hand pose keypoints. Therefore, we used combinations of body pose (🧍), hand poses (✋) and eye gazes (👁) to form the input as they are computationally more efficient than deep visual features. We additionally compared the performance when using video features from VideoMAE as input.

**Baselines and Metrics.** We consider state-of-art pose estimation models proposed in DLC2Action [32] to perform action segmentation. The toolbox adapted state-of-the-art models for

RGB-based action segmentation tasks (Breakfast [33] and 50Salads [76]), such as MS-TCN++ [39], EDTCN [35], and C2F-TCN [75], to work directly on pose estimation data (vs deep visual features): MS-TCN3 and C2F-Transformer. We used kinematic features and VideoMAE [83] features as input to the models (Supp. Sec. F.6.3). Each action is evaluated separately using standard metrics in action segmentation (Frame-wise F1, F1@50, edit distance) and ultimately averaged over action groups. All models were trained and evaluated using the DLC2Action toolbox [32].

**Results**. We observed that the benchmark is challenging for current action segmentation algorithms (Table 4 and Supp. Table F.4). Exo-Body performs similarly to Exo-Hand with a slight improvement for Exo-Hand albeit for different behaviors. We note that video information provided a boost in performance. These baselines showed an F1-score of 35.2% for verbs and 35.0% for nouns, highlighting significant potential for future advancements.

Table 4: **F1 scores for action segmentation benchmark.** 🧍: 3D body pose, 🖐: 3D hand pose, 👁: eye gaze, 📷: egocentric view. *models modified to use pose as input data instead of image features.

| | Verbs | | | | | | Nouns | | | | | | Activity | | | | | |
|---|---|---|---|---|---|---|---|---|---|---|---|---|---|---|---|---|---|---|
| | 🧍 | 🖐 | 👁 | 🧍🖐 | 🧍🖐👁 | 🧍🖐👁📷 | 🧍 | 🖐 | 👁 | 🧍🖐 | 🧍🖐👁 | 🧍🖐👁📷 | 🧍 | 🖐 | 👁 | 🧍🖐 | 🧍🖐👁 | 🧍🖐👁📷 |
| MS-TCN3 | 18.1 | 20.2 | 11.7 | 20.9 | 21.1 | 30.1 | 10.6 | 13.4 | 7.6 | 15.6 | 11.3 | 31.2 | 51.8 | 58.6 | 31.9 | 54.4 | 58.5 | **72.9** |
| C2F-TCN* | 18.8 | 20.1 | 12.2 | 22.1 | 22.2 | 34.6 | 12.0 | 14.3 | 7.9 | 16.1 | 10.8 | **35.2** | 54.5 | 55.4 | 41.3 | 61.8 | 61.2 | 72.2 |
| C2F-Transf. | 19.9 | 22.4 | 13.1 | 22.8 | 22.2 | **35.0** | 11.1 | 12.9 | 7.8 | 13.4 | 9.2 | 29.0 | 51.2 | 56.9 | 38.8 | 62.1 | 59.9 | 70.5 |
| EDTCN* | 19.6 | 23.0 | 11.9 | 22.1 | 25.2 | 34.3 | 11.9 | 11.2 | 7.1 | 12.3 | 11.9 | 24.3 | 49.0 | 53.5 | 32.0 | 53.1 | 54.2 | 71.0 |

### 4.4 Situated full-body motion generation benchmark

**Rationale.** With the diverse actions and motions in EPFL-Smart-Kitchen-30, our motion generation benchmark has three key innovations over the commonly used KIT [64] and HumanML3D [26] benchmarks. Existing motion generation benchmarks mainly focus on broad daily activities, sports, and dance, whereas the EPFL-Smart-Kitchen-30 contains hundreds of fine-grained behaviors. We extend motion generation beyond the body to include hands and eye gaze, defining it as full-body motion generation. Additionally, we provide egocentric visual features alongside action text as the condition, allowing situated motion generation.

**Pre-processing, Baselines and Metrics** To achieve robust full-body motion representation, we combine joint locations and angles for the body, hands, and eye gaze, resulting in a 327-dimensional redundant motion representation. We process fine-grained action text with linguistic tags and extract egocentric visual features with CLIP's text and image encoders [66]. As baselines, we adapt three strong motion generation models, T2M-GPT [99], MARDM-SiT [55] and MoMask [27], training them with verb-noun pairs and verb-only prompts (Supp. Sec. F.7). Like in HumanML3D [26], we train the quantitative evaluator on EPFL-Smart-Kitchen-30 to measure the R-Precision top-1 to top-3 (T1 to T3), multimodal distance (MMd), Fréchet Inception Distance (FID), Diversity (DIV), and Multimodality (MM).

**Results.** Our qualitative analysis (from MoMask [27]) revealed compelling examples of successfully generated full body motion sequences that accurately represent natural cooking behaviors (Figure 5). Quantitatively, we found that just like on HumanML3D[26], MoMask [27] consistently outperforms T2M-GPT [99] and MARDM-SiT [55] in different settings and metrics (Table 5), possibly due to the hierarchy and quantized tokenization way working better for the high-dimensional properties of the

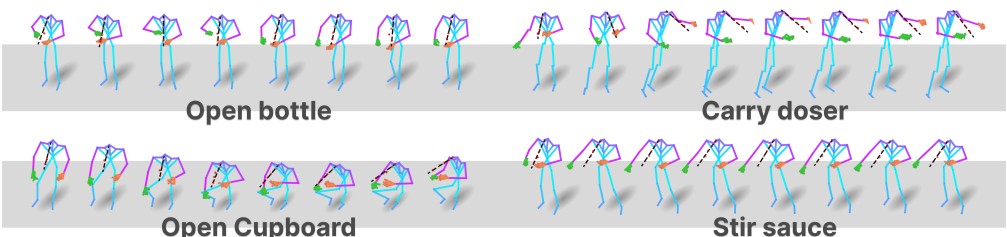

Figure 5: **Qualitative motion generation samples** from MoMask trained with Verb-Noun action pairs. The model creates realistic hand, body and eye gaze (dashed line).

full-body representation. Furthermore, when conditioning on the egocentric view, the model was able to find some clues to minimize the global distribution, and thus make the FID score lower.

Table 5: **Full-body motion generation results**. Models trained and evaluated on the EPFL-Smart-Kitchen-30. FID:Fréchet Inception Distance, DIV:Diversity, MM:Multimodality, MMd:Multimodal distance.

| Condition | Vocab. | Model | T1 ↑ | T2 ↑ | T3 ↑ | FID ↓ | DIV ↑ | MM ↑ | MMd ↓ |
|---|---|---|---|---|---|---|---|---|---|
| Text | Verb | T2M-GPT | 0.254 | 0.432 | 0.567 | 2.640 | 7.288 | 1.555 | 3.379 |
| | | MARDM-SiT | 0.262 | 0.473 | 0.637 | 2.242 | 7.590 | 1.849 | 4.155 |
| | | Momask | 0.306 | 0.508 | 0.652 | 3.124 | **8.347** | **2.940** | **2.048** |
| | Actions | T2M-GPT | 0.271 | 0.434 | 0.542 | 2.378 | 7.031 | 0.852 | 4.015 |
| | | MARDM-SiT | 0.320 | 0.548 | 0.650 | 1.230 | 7.704 | 1.427 | 4.103 |
| | | Momask | **0.372** | **0.566** | **0.683** | 0.930 | 7.859 | 1.641 | 3.407 |
| Text-Image | Verb | T2M-GPT | 0.174 | 0.295 | 0.387 | 2.415 | 6.558 | 0.822 | 4.524 |
| | | MARDM-SiT | 0.187 | 0.355 | 0.466 | 1.498 | 6.579 | 1.581 | 4.691 |
| | | Momask | 0.197 | 0.333 | 0.436 | 0.858 | 6.696 | 1.665 | 4.121 |
| | Actions | T2M-GPT | 0.243 | 0.398 | 0.506 | 1.982 | 6.633 | 0.717 | 4.125 |
| | | MARDM-SiT | 0.255 | 0.469 | 0.597 | 0.917 | 7.176 | 1.322 | 4.725 |
| | | Momask | 0.276 | 0.441 | 0.552 | **0.627** | 7.141 | 1.554 | 3.937 |

## 5 Conclusion, future work and impact

We collected 30 hours of RGB-D video from ten synchronized views, 3D pose data, and hierarchical action annotations in a calibrated kitchen environment. All participant data has been anonymized to protect privacy (Table 6). The dataset's multimodal nature and fine-grained annotations enable analysis of complex behavioral patterns, object interactions, and visual attention mechanisms during goal-directed activities (Figure 2C). Our work complements recent large-scale datasets such as EPIC-KITCHENS-100, EgoExo4D, and Humans in Kitchens [14, 79, 24] by providing integrated multimodal data streams within a controlled environment. While these datasets excel in scale and environmental diversity, EPFL-Smart-Kitchen-30 offers synchronized multi-view video, pose estimation, and eye tracking data that enables new research directions in multimodal behavior understanding.

The four benchmarks we propose—action recognition, action segmentation, motion generation, and video question answering—demonstrate both the potential and current limitations of existing models on fine-grained behavioral tasks. Future work could leverage additional modalities in our dataset (IMU data, depth information) and develop models that more effectively integrate multiple data streams for robust behavior understanding. Additionally, we are collecting data from older and non-healthy participants (stroke and amputee patients) for future release, aiming to improve treatments for subjects with neurological disorders [56]. This will also increase the demographic representation of our dataset. Overall, we share multi-view, multimodal action understanding, modeling, and video question answering benchmarks to leverage the potential of multi-modality for improving action understanding and to fuel foundation models. This is particularly interesting for emerging multi-modal models [48, 19, 60, 95].

**Acknowledgments:** We thank members of the Mathis Group for Computational Neuroscience & AI (EPFL) for their feedback throughout the project. This work was funded by EPFL, Swiss SNF grant (320030-227871), Microsoft Swiss Joint Research Center, and a Boehringer Ingelheim Fonds PhD stipend (H.Q.). We are grateful to the Brain Mind Institute for providing funds for hardware and to the Neuro-X Institute for providing funds for services.

**Dataset and Code Release:** Please check `https://amathislab.github.io/EPFL-Smart-Kitchen` for the latest updates.

Table 6: **Data and Code Availability**

| Resource | Location |
|---|---|
| Code Repository | `https://github.com/amathislab/EPFL-Smart-Kitchen` |
| Dataset (Collected Data) | `https://zenodo.org/records/15535461` |
| Annotations (Pose & Behavior) | `https://zenodo.org/records/15551913` |
| Benchmark dataset and checkpoints | `https://huggingface.co/collections/amathislab/esk-benchmarks` |

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
