# OpenReview forum: "EPFL-Smart-Kitchen: An Ego-Exo Multi-Modal Dataset for Challenging Action and Motion Understanding in Video-Language Models"
_NeurIPS.cc/2025/Datasets_and_Benchmarks_Track — NeurIPS 2025 Datasets and Benchmarks Track poster_

### Official Review · Reviewer_KvEi · 2025-06-25

**Rating:** 5
**Confidence:** 4

**Summary:**

This paper introduces X-Smart-Kitchen-30, a new large-scale, multi-modal human behavior dataset collected in a complex kitchen environment. The dataset includes a wide range of modalities: RGB video, depth, IMU, eye gaze, and 3D skeletal data (including both body and fine-grained hand joints). Notably, it features 4 exocentric views for capturing global human actions, 5 exocentric views for detailed local actions, and 1 egocentric view for a first-person perspective. Furthermore, the authors propose four novel benchmarks designed to advance fine-grained behavior understanding and generation: VLM-based VQA, action recognition, action segmentation, and full-body motion generation. Overall, this dataset is a valuable resource likely to inspire and support diverse research directions in multi-modal human behavior modeling.

**Additional Feedback:**

1. The inclusion of IMU data is potentially valuable, but its role within the proposed benchmarks is unclear. The paper does not sufficiently explain how this modality can contribute to the proposed four tasks. Clarifying the motivation for collecting this non-intrusive data, along with examples of its potential usage, would improve the dataset’s completeness and relevance.
2. The format and structure of the eye gaze data are not clearly described. Providing more details would be helpful for researchers aiming to utilize this modality effectively.

**Dataset Code Accessibility:**

Yes

**Dataset Code Comments:**

The dataset URL is publicly available, and the supplementary material is thorough and comprehensive, providing detailed information about the dataset’s structure and content. This level of documentation is helpful for researchers aiming to understand and utilize the dataset effectively.

**Ethical Considerations:**

No, there are no or only very minor ethics concerns

**Final Justification:**

The authors have adequately addressed my concerns. As my original rating was already “accept,” I will maintain this score.

**Limitations Weaknesses:**

1. Some modalities, such as IMU data, are underutilized in the paper. Their potential applications in human-centric tasks are not sufficiently discussed. Similarly, the format of the eye gaze data is not clearly described, and its role or necessity in downstream tasks is not well justified. Including more discussion or experimental exploration of these modalities would strengthen the paper and improve the practical usability of the dataset.
2. The dataset is collected entirely within a single kitchen environment, which may limit its generalizability across different real-world settings.

**Strengths Contributions:**

1. The dataset is large-scale, richly annotated, and captured in a realistic and dynamic kitchen setting. In particular, the inclusion of fine-grained hand pose and eye gaze data provides substantial value. Additionally, the availability of egocentric views for motion generation is particularly commendable and opens up new research avenues.
2. The paper is clearly written, logically structured, and visually well-presented. Figures are informative and visually engaging. The hierarchical action annotation is well-conceived, and the dataset’s statistical analysis is both detailed and convincing. Experimental results across the four benchmarks are comprehensive and robust.
3. This dataset effectively integrates the strengths of prior works while introducing fresh and promising insights, especially in the proposed benchmarks. The full-body motion generation conditioned on eye gaze is particularly novel and could have a significant impact on the human motion generation community.

---

> ### Author Rebuttal · Authors · 2025-07-31
>
> We thank the reviewers for their thoughtful comments and the effort spent reviewing our paper. We appreciate that the reviewer agreed with the value of our multi-modal data and the promising insight of our benchmarks.
>
> > 1. “Some modalities, such as IMU data, are underutilized in the paper. Their potential applications in human-centric tasks are not sufficiently discussed.”
>
> The IMU information is shared to the community to use for their convenience. The advantage of the IMU information in particular for the knife and the spatula, over the wrist pose estimation, is the higher sampling frequency (100Hz > 30fps).
> This increased temporal resolution can give insights in precise event timings, for instance when the knife touches the cutting board in a cut, or when the spatula is being grabbed.
> In addition, the profile of the IMU patterns can give information about the type of object being interacted with through its oscillations. IMU data on knife and spatula can also give clues about the hand motions during interaction, providing a possible low-cost way of hand motion perception or analysis.  We will add potential applications of the collected IMU data in the discussion.
>
> > 1. “The format of the eye gaze data is not clearly described, and its role or necessity in downstream tasks is not well justified. Including more discussion or experimental exploration of these modalities would strengthen the paper and improve the practical usability of the dataset.”
>
> To recapitulate, the eye gaze is given as a set of keypoints sampled from a measured direction at a constant distance (from 0m to 0.9m). It is used in the action recognition benchmark, the action segmentation benchmark and the motion generation benchmark.  As we can see, concatenating the eye gaze typically does not improve the performance of the models in the action recognition and the action segmentation benchmarks.
> Since gaze nature is more anticipatory (Mennie et al. [1]), it might better predict the next action rather than the current action and therefore confuse the current action prediction. In the motion generation benchmark, this is a novelty that can be used in the animation of more realistic agents and would also be beneficial for embodied AI.
>
>
> > 2. “The dataset is collected entirely within a single kitchen environment, which may limit its generalizability across different real-world settings.”
>
> We focus on capturing high-quality multi-view video and continuous full body motion data to support multi-modal action understanding. Similar to other complex datasets with hand motions captured such as Assemblyhands [2], this commitment to data fidelity naturally limits the diversity of captured scenarios. As in many other areas in AI, generalizability requires the community to contribute together.
>
> Upon acceptance, we will revise our paper and add the necessary experiments used in the rebuttal to the camera-ready paper. We hope that our responses have sufficiently addressed all comments.
>
>
> ## References
>
> [1] Mennie, Neil, Mary Hayhoe, and Brian Sullivan. "Look-ahead fixations: anticipatory eye movements in natural tasks." Experimental brain research 179.3 (2007): 427-442.
>
> [2] Ohkawa, Takehiko, et al. "Assemblyhands: Towards egocentric activity understanding via 3d hand pose estimation." Proceedings of the IEEE/CVF conference on computer vision and pattern recognition. 2023.

---

> > ### Author Response · Authors · 2025-08-04
> > **Kind Reminder: Seeking Reviewer Feedback for Author/Reviewer Discussion Phase**
> >
> > Dear reviewer KvEi,
> >
> > We hope this message finds you well. We would like to express our gratitude for the time and effort that you have dedicated to the review process of our submission.
> >
> > Furthermore, we are writing to kindly remind you that the rebuttal period is coming to a close. Your feedback and evaluation are crucial to the progress of our work, and we value your expertise and insights immensely.
> >
> > During rebuttal, we hope to have addressed all the queries and concerns you raised regarding our submission. Each point has been carefully considered, and we have provided detailed responses.
> >
> > We are eager to engage in further discussions with you on the responses and any other aspects of our study. Your feedback plays a pivotal role in shaping the direction of our research, and we are keen to hear your thoughts and suggestions.
> >
> > Thank you for your consideration.
> >
> > Best regards,
> >
> > Authors of submission #441

---

> > ### Comment · Reviewer_KvEi · 2025-08-05
> > **Rebuttal Response**
> >
> > Thank you to the authors for their detailed responses. All of my concerns have been addressed. I will maintain my original score.

---

> > > ### Author Response · Authors · 2025-08-07
> > >
> > > Thank you for your time and effort in reviewing the paper and the rebuttal, as well as for maintaining your positive evaluation. We will also make sure to include your suggestions in the final version.

---

### Official Review · Reviewer_TcCr · 2025-06-26

**Rating:** 5
**Confidence:** 4

**Summary:**

This paper introduces the X-Smart-Kitchen-30 dataset, a multi-modal cooking dataset captured via 9 RGB-D cameras, IMUs, and a HoloLens 2 headset, featuring 29.7 hours of data from 16 subjects cooking 4 recipes with dense annotations (avg 33.78 action segments per minute). It presents four benchmarks: vision-language, semantic text-to-motion generation, multi-modal action recognition, and pose-based action segmentation. The dataset addresses fragmentation in behavioral analysis by integrating full-body/hand/eye tracking and multi-view video, enabling fine-grained action understanding and advancing models for human behavior research.

**Dataset Code Accessibility:**

Yes

**Ethical Considerations:**

Yes, there are ethics concerns that require attention by the authors

**Final Justification:**

I stand by my judgment that this paper deserves acceptance.

**Limitations Weaknesses:**

1. Benchmarks focus on cooking tasks, potentially limiting generalizability to other domains (e.g., industrial assembly).
2. Current VLMs struggle with kinematic inference (e.g., 42% accuracy in Lemonade’s Kinematics subcategory, Fig. 4C), indicating limited model readiness.
3. Rare actions (≤3 instances) are excluded from benchmarks (Sec. 4), potentially biasing results toward common behaviors.

**Strengths Contributions:**

1. Addresses fragmentation in behavioral analysis by integrating rich multi-modal sensors (full-body/hand/eye tracking and multi-view video), enabling fine-grained action understanding.
2. Well-established benchmarks (vision-language, text-to-motion generation, action recognition, pose-based segmentation) to advance behavior modeling.
3. Effectively justifies gaps in current datasets and highlights the dataset’s utility for advancing multi-modal models and human behavior research.
4. Well-structured wrting with clear figures.

---

> ### Author Rebuttal · Authors · 2025-07-31
>
> We thank the reviewers for their thoughtful comments and the effort spent reviewing our paper. We appreciate that the reviewer recognizes our dataset’s contribution compared to others and likes our benchmarks.
>
> >1. “Benchmarks focus on cooking tasks, potentially limiting generalizability to other domains (e.g., industrial assembly)“
>
> As we discussed in Table 1, our dataset focuses on both video-based and motion-based action understanding. Therefore, we aim to have both muti-view visual information as well as accurate and continuous full body motion data. This limits our dataset scenario set similar to other relevant datasets with hand motions annotated (Assemblyhands [1]) that aim to capture comparable information. As in many other areas in AI, generalizability requires the community to contribute together. Meanwhile, the cooking scenario is an excellent environment that contains various and complex hand movements, which can also generalize to other domains.
>
> >2. “Current VLMs struggle with kinematic inference (e.g., 42% accuracy in Lemonade’s Kinematics subcategory, Fig. 4C), indicating limited model readiness.”
>
> We use our dataset to formulate a zero-shot VQA benchmark in order to test the motion and action understanding ability of the current VLMs. The fact that current VLMs are struggling with kinematic inference shows the value of our dataset as well as the Lemonade benchmark. We hope researchers will use our benchmark to discover limitations and improve their current models.
>
> >3. "Rare actions (≤3 instances) are excluded from benchmarks (Sec. 4), potentially biasing results toward common behaviors."
>
> We remove the actions with less than 3 instances to make sure that at least one instance appears in the train, validation and test splits. Long tail distribution is a common problem. As we showed in Table 2, models will perform better on more common classes. Here, we test one method (Perrett, Toby, et al [2]) that tries to deal with long-tail distribution. It improves the Tail Top 1 class Accuracy to 23.77% while decreasing the Head Top 1 class Accuracy to 35.62%. We encourage the community to discover better algorithms to handle long tail distribution.
>
> Upon acceptance, we will revise our paper and add the necessary experiments used in the rebuttal to the camera-ready paper. We hope that our responses have sufficiently addressed all comments.
>
>
> ## References
>
> [1] Ohkawa, Takehiko, et al. "Assemblyhands: Towards egocentric activity understanding via 3d hand pose estimation." Proceedings of the IEEE/CVF conference on computer vision and pattern recognition. 2023.
>
> [2] Perrett, Toby, et al. "Use your head: Improving long-tail video recognition." Proceedings of the IEEE/CVF conference on computer vision and pattern recognition. 2023.

---

> > ### Author Response · Authors · 2025-08-04
> > **Kind Reminder: Seeking Reviewer Feedback for Author/Reviewer Discussion Phase**
> >
> > Dear reviewer TcCr,
> >
> > We hope this message finds you well. We would like to express our gratitude for the time and effort that you have dedicated to the review process of our submission.
> >
> > Furthermore, we are writing to kindly remind you that the rebuttal period is coming to a close. Your feedback and evaluation are crucial to the progress of our work, and we value your expertise and insights immensely.
> >
> > During rebuttal, we hope to have addressed all the queries and concerns you raised regarding our submission. Each point has been carefully considered, and we have provided detailed responses.
> >
> > We are eager to engage in further discussions with you on the responses and any other aspects of our study. Your feedback plays a pivotal role in shaping the direction of our research, and we are keen to hear your thoughts and suggestions.
> >
> > Thank you for your consideration.
> >
> > Best regards,
> >
> > Authors of submission #441

---

> > > ### Author Response · Authors · 2025-08-07
> > >
> > > We're deeply grateful for your positive assessment. Your recognition of our work's technical quality and potential impact means a great deal to us. Your observation about VLM limitations indeed validates Lemonade's value as a novel benchmark to advance VLMs - it reveals important gaps that need addressing.

---

> ### Comment · Area_Chair_bLDD · 2025-08-08
>
> Dear Reviewer TcCr,
>
> Thank you again for reviewing the paper. Could you please check the author feedback? Please also be reminded that the “Final Justification” should be completed by August 13.
>
> Best,
> AC

---

### Official Review · Reviewer_tnRW · 2025-07-02

**Rating:** 4
**Confidence:** 4

**Summary:**

The authors introduce X-Smart-Kitchen-30, a 29.7-hour, multi-view, multi-modal dataset of 16 participants cooking four recipes in a realistic kitchen setting, captured simultaneously by nine static RGB-D cameras, a head-mounted HoloLens 2 (egocentric RGB-D and eye gaze) and eight body-worn IMUs. Synchronized exocentric and egocentric video, inertial measurements, eye‐tracking data and 3D hand/body kinematics yield dense annotations—an average of 33.8 action segments per minute—across diverse subjects. To validate its utility, the paper defines four benchmarks—question-answering benchmark, semantic text-to-motion generation, multi-modal action recognition and pose-based action segmentation—showing that integrating IMU and kinematic streams with visual inputs improves recognition accuracy over vision-only models. X-Smart-Kitchen-30 thus provides a richly annotated, ecologically valid platform for advancing behavior understanding, motion synthesis and embodied AI.

**Additional Feedback:**

See Weaknesses

**Dataset Code Accessibility:**

Yes

**Ethical Considerations:**

No, there are no or only very minor ethics concerns

**Final Justification:**

Although I still don’t fully agree with the author’s responses, the author put a lot of effort into the discussion and added experiments to substantiate their claims, so I will change my score from “Borderline reject” to “Borderline accept.”

**Limitations Weaknesses:**

Overall, I appreciate the effort involved in assembling a richly annotated kitchen dataset, but in my view X-Smart-Kitchen-30 overlaps substantially with existing resources such as EgoExo4D—both offer synchronized egocentric and third-person video, action labels, and multi-modal streams—and yet it does so at a much smaller scale (single “lab” kitchen, only four recipes). Because larger, more diverse datasets with broader benchmarks are already available, this new collection, while valuable in its own right, is unlikely to attract widespread adoption or drive significant progress in the community.
1. Limited environmental diversity. All recordings take place in a single “lab” kitchen, whereas datasets like EPIC-Kitchens and EgoExo4D include real-home or multiple environments, yielding more naturalistic motions and broader context coverage .
2. Small recipe set. The dataset covers only four pre-selected recipes, compared to 10 in Breakfast and 14 in EgoExo4D, constraining the range of procedural variations and reducing comparability across more diverse cooking tasks .
3. Overlap with EgoExo4D. EgoExo4D already provides extensive first- and third-person video across multiple environments and activities; while X-Smart-Kitchen-30 adds depth and IMU streams, its smaller scale limits its novelty. Annotating EgoExo4D footage might have yielded greater impact.
4. Sparse and under-utilized IMUs. IMU sensors are mounted on only six doors, a knife and a spatula, and are used solely as input modalities rather than enabling dedicated inertial-based benchmarks, underplaying one of the dataset’s claimed innovations.



[1] Grauman R et al. EgoExo4D: Understanding skilled human activity from first-and third-person perspectives. CVPR 2024.
[2] Damen D et al. Scaling egocentric vision: The EPIC-Kitchens dataset. ECCV 2018.

**Strengths Contributions:**

1. The manuscript is written in a lucid, highly accessible style, and provides a thorough, step-by-step introduction to the new dataset.

2. By combining multiple perspectives (static multi-view cameras, egocentric HoloLens feeds, eye-tracking, object-mounted IMUs, etc.) across four distinct recipes, the dataset delivers rich, long-duration sequences ideal for in-depth semantic analysis.

---

> ### Author Rebuttal · Authors · 2025-07-31
>
> We thank the reviewers for their thoughtful comments and the effort spent reviewing our paper. We appreciate that the reviewer saw the potential of the dataset in terms of semantic analysis.
>
> >1. “Limited environmental diversity. All recordings take place in a single “lab” kitchen, whereas datasets like EPIC-Kitchens and EgoExo4D include real-home or multiple environments, yielding more naturalistic motions and broader context coverage .”
>
> A platform that can precisely record both body and hand motions limits environmental diversity.
>
> Understanding human actions from video and motion perspectives are both important but mostly separate in previous studies, due to the difficulty of capturing both modalities at the same time. As we highlighted in our Table 1, recently there are datasets trying to focus on both video-based and motion-based action understanding. Our dataset falls into this direction, and has a very different focus in comparison to EPIC-Kitchens and EgoExo4D, which are mainly video-oriented.
>
> Although EgoExo4D has partial 3D pose annotations, those annotations are discontinuous and are only keypoint-based. Capturing parametric models for rich and accurate motion representation over long timescales on Ego-Exo4D and other environment-diverse datasets will be very hard due to the limited number of cameras in their setups. Therefore, Ego-Exo4D mainly has video-relevant benchmarks when they are related to action understanding.
>
> Among the motion-based action understanding, hand motion is one of the most challenging modalities to capture but represents a core of human activity. Capturing **complex long-lasting hand motions is one of our main focuses** and differs substantially from Ego-Exo4D since more complicated camera and sensor setups are needed. Other excellent and existing datasets (Assemblyhands [1], UmeTrack [2], HOT3D [3]) with hand-motion annotations are constrained in smaller workspaces and environments. In summary, EgoExo4d does not have more "naturalistic motions" compared to ours.
>
> >2. “Small recipe set. The dataset covers only four pre-selected recipes, compared to 10 in Breakfast and 14 in EgoExo4D, constraining the range of procedural variations and reducing comparability across more diverse cooking tasks .”
>
> The choice of the recipe set was a deliberate design choice that prioritizes depth of behavioral analysis over procedural breadth. Our primary goal is not to capture a wide variety of cooking tasks, but rather to build a rich and detailed representation of fundamental human actions and movements with the following two facts.
> 1) Two of our recipes (Omelette and Ratatouille) are cooked by all of the subjects, which covers the various within-recipe actions and motions.
> 2) One recipe (Ratatouille) is cooked twice by the subjects, which would cause action and motion adaptation and differences (Charness et al. [4]).
>
> With the above two facts, we believe our dataset can be an excellent complementary to the existing datasets that cares more about recipe diversity.
>
> Meanwhile, the unconstrained nature of our protocol ensures that we capture significant variability in how actions like "cutting" or "stirring" are performed across numerous participants. We show the richness of hand motions in our dataset by reporting a high number of their DoFs (Figure 2C). The repetition of these actions is crucial for our objective: by collecting many instances of the same core behaviors, we provide the dense data necessary to train robust models that can generalize on these specific, fine-grained actions. This approach intentionally mirrors controlled experimental designs in science, where repetition in a focused setting enables a detailed and rigorous study of specific phenomena. We believe this design makes our dataset uniquely valuable for the granular analysis of human behavior and positions it to contribute meaningfully via AI4Science to Neuroscience, Psychology, etc. .
>
> >3. “Overlap with EgoExo4D. EgoExo4D already provides extensive first- and third-person video across multiple environments and activities; while X-Smart-Kitchen-30 adds depth and IMU streams, its smaller scale limits its novelty. Annotating EgoExo4D footage might have yielded greater impact.”
>
> As we discussed in the first comment, using fewer cameras and sensors in set-ups makes accurate and continuous motion data prediction, especially hand motions, very challenging. Although EgoExo4D has partial 3D pose annotations, it is still video-focused and is not suitable for video-motion focused action understanding. In contrast, we record from 10 different angles while including global and local views to well assess both body and hand pose estimation in an unconstrained environment. Therefore, we argue that our dataset provides excellent complementation to the existing datasets.
>
>
> >4. “Sparse and under-utilized IMUs. IMU sensors are mounted on only six doors, a knife and a spatula, and are used solely as input modalities rather than enabling dedicated inertial-based benchmarks, underplaying one of the dataset’s claimed innovations.”
>
> We appreciate the reviewer's perspective on the IMU data. One key contribution of this dataset is to provide this very data to the community for dedicated future research. Providing data that isn’t dedicated to a benchmark is also common in other datasets such as EgoExo4D and HoloAssist [5].
>
> The IMU data, especially from the knife and spatula, offers a significant increase in temporal resolution compared to video. This high-frequency data contains rich signals crucial for analyzing complex, fine-grained manipulations, which we believe is a valuable asset for the research community. Specifically, the IMU information contains precise timings to when the knife touches the cutting board, or when the spatula is being grabbed. In addition, the profile of the IMU patterns can give information about the object being interacted with through its oscillations. Finally, IMU data on knife and spatula can also reflect hand motions during the interaction, which could inspire low-cost motion perception or analysis for the research community. We will edit the manuscript to emphasize on the advantages that IMU data can provide.
>
>
> Upon acceptance, we will revise our paper and add the necessary experiments used in the rebuttal to the camera-ready paper. We hope that our responses have sufficiently addressed all comments.
>
>
> ## References
>
> [1] Ohkawa, Takehiko, et al. "Assemblyhands: Towards egocentric activity understanding via 3d hand pose estimation." Proceedings of the IEEE/CVF conference on computer vision and pattern recognition. 2023.
>
> [2] Han, Shangchen, et al. "UmeTrack: Unified multi-view end-to-end hand tracking for VR." SIGGRAPH Asia 2022 conference papers. 2022.
>
> [3] Banerjee, Prithviraj, et al. "Hot3d: Hand and object tracking in 3d from egocentric multi-view videos." Proceedings of the Computer Vision and Pattern Recognition Conference. 2025.
>
> [4] Charness, Gary, Uri Gneezy, and Michael A. Kuhn. "Experimental methods: Between-subject and within-subject design." Journal of economic behavior & organization 81.1 (2012): 1-8.
>
> [5] Wang, Xin, et al. "Holoassist: an egocentric human interaction dataset for interactive ai assistants in the real world." Proceedings of the IEEE/CVF International Conference on Computer Vision. 2023.

---

> > ### Author Response · Authors · 2025-08-04
> > **Kind Reminder: Seeking Reviewer Feedback for Author/Reviewer Discussion Phase**
> >
> > Dear reviewer tnRW,
> >
> > We hope this message finds you well. We would like to express our gratitude for the time and effort that you have dedicated to the review process of our submission.
> >
> > Furthermore, we are writing to kindly remind you that the rebuttal period is coming to a close. Your feedback and evaluation are crucial to the progress of our work, and we value your expertise and insights immensely.
> >
> > During rebuttal, we hope to have addressed all the queries and concerns you raised regarding our submission. Each point has been carefully considered, and we have provided detailed responses.
> >
> > We are eager to engage in further discussions with you on the responses and any other aspects of our study. Your feedback plays a pivotal role in shaping the direction of our research, and we are keen to hear your thoughts and suggestions.
> >
> > Thank you for your consideration.
> >
> > Best regards,
> >
> > Authors of submission #441

---

> > > ### Author Response · Authors · 2025-08-04
> > >
> > > We appreciate your questions regarding EgoExo4D, which further motivates to clarify our dataset's contributions:
> > >
> > > **Complementary Focus**: While EgoExo4D provides broad environmental coverage for skill understanding, X-Smart-Kitchen-30 addresses the orthogonal challenge of accurate motion capture for behavior modeling. These represent different research priorities within the broader human understanding ecosystem.
> > >
> > > **Technical Capabilities**: Our 10-camera setup enables continuous parametric pose models and dense hand tracking throughout 29.7 hours—capabilities that EgoExo4D's sparser camera configuration cannot achieve. This precision enables us to propose benchmarks for multimodal motion synthesis, and action segmentation. Our *text-to-motion generation benchmark* enables training models that can synthesize realistic cooking behaviors conditioned on natural language and visual context. It will also enable detailed future kinematic analysis
> > >
> > > **Study design**: Our controlled recipe repetition follows established experimental design principles (in behavioral science), enabling statistical analysis of behavioral variations across individuals and sessions. This controlled depth complements EgoExo4D's environmental breadth.
> > >
> > > **Bridge Vision-Motion Gap**: X-Smart-Kitchen-30 uniquely combines high-frequency multi-modal streams (100Hz IMU, 30fps parametric poses, multi-view RGBD) in a unified framework, addressing the current fragmentation between video and motion research communities. This enables researchers to leverage both our precise motion data and the environmental diversity of EPIC-Kitchens, EgoExo4D, and other datasets for comprehensive behavior understanding.

---

> > ### Comment · Reviewer_tnRW · 2025-08-05
> >
> > Thanks to the authors for their detailed responses addressing my concerns about the small recipe set. I still have two remaining issues regarding the EgoExo4D dataset and its IMU data:
> >
> > 1. I appreciate the importance of capturing human actions from both video and motion perspectives. However, in EgoExo4D the “motion” signals are produced by a pose-estimation algorithm rather than being recorded directly from inertial sensors. The same algorithm could just as easily be applied to existing multi-view video datasets to generate comparable motion streams. This raises questions about what unique contribution EgoExo4D offers on the motion side.
> >    On a related note, I’m not convinced that having fewer cameras and sensors inherently limits action understanding; in fact, a lighter setup may better reflect real-world constraints, where extensive rigs are impractical. Yet the paper stops short of examining how varying the number of devices actually affects model performance.
> >
> > 2. I applaud the inclusion of IMU recordings, but if they are to be a key asset, the manuscript needs a more thorough investigation of their impact. As it stands, it is unclear how the IMU streams are integrated into the proposed method or why they improve action recognition. A detailed ablation or case study demonstrating what the IMU data contribute—and under which circumstances—would greatly strengthen the paper’s claims.

---

> > > ### Author Response · Authors · 2025-08-07
> > >
> > > We thank the reviewer for engaging in further discussion. Please consider the responses below related to the two remaining issues.
> > >
> > > 1.We thank the reviewer for agreeing to the importance of understanding human actions from both video and motion perspectives, which is the key contribution of our dataset.
> > > Regarding the concern of applying our motion estimation method to other multi-view video datasets (such as EgoExo4D), we argue that **accurate and continuous 3D human motions, especially 3D hand motions, require both  a good motion estimation method and to carefully set sensory setups**. As we discussed in Sec. 3.3, for the exocentric views, we carefully installed four cameras to capture global body information and five cameras to mainly capture local information targeted for hand capture. We are currently running an ablation analysis to show how the pose estimation performance depends on the number of cameras.
> > >
> > > In comparison, as can be seen in **Fig.1 and Fig. 8 of EgoExo4D’s manuscript**, its exocentric views, which have 3-4 views, focus only on global body information,  making it very hard to estimate 3D hand poses. This could be one of the main reasons that would explain why EgoExo4D only shares partial frame-based body/hand pose annotations;  they likely could only annotate when the keypoints are visible on at least two views. In summary, we agree that lighter setups could better reflect real-world constraints (like EgoExo4D) but fewer cameras and sensors cannot infer reliable motions especially for the hands, and that is the key reason why understanding human actions from video and motion perspectives are separate. Indeed, this opens interesting future directions.
> > >
> > > Indeed, for action recognition we already show clear evidence for this. Namely, the best multimodal model for action recognition in our benchmark includes the “best” hand view  (Table 2). This improves performance by  6.3% when trained from the pretrained model. Our dataset tries to fill this gap by carefully designing our sensory setup.
> > >
> > > 2.We appreciate your question regarding the IMU’s potential. Therefore, we conduct a case study to demonstrate IMU’s contribution as follows:
> > >
> > > **Intuition**: The IMU sensors record information on the interacted object at a higher frequency than the motion capture system (100Hz vs 30Hz). Can the sensor attached to the knife capture differences in object textures in cutting events? In particular, do higher frequency signal patterns contain “cut object” signatures?
> > >
> > > **Method**: To test this hypothesis, we compare the power spectrum densities of the knife IMU sensor and the dominant participants’wrist 3D estimated pose during cutting sequences of different objects. We then run statistical tests that compare the total spectral power of each sequence grouped by cut object (n=21 objects), in different frequency bands (0.1Hz-4Hz, 4Hz-6Hz, 6Hz-8Hz, 8Hz-100Hz). Specifically, we run a Wilcoxon signed-rank test (non-parametric pairwise test) with Benjamini-Hochberg FDR correction on distribution of objects per frequency band after outlier removal.
> > >
> > > **Results**: We found that the frequency profile of the knife IMU sensor during cutting events can significantly distinguish between 38 pairs of objects while the wrist signal can only distinguish 10 pairs of objects (p-value < 0.05; CLES > 0.7 and CLES >0.3). The high frequency profile might refer to specific oscillations specific to the object texture or the force applied with respect to these specific objects.

---

> > > > ### Author Response · Authors · 2025-08-07
> > > >
> > > > Furthermore, we appreciate the reviewer’s comments about the influence of different numbers of devices and conduct an ablation that combines the egocentric view and three/four global camera views for motion estimation, which is comparable to EgoExo4D’s setup. The results, showing pose estimation error (MPJPE=Mean per joint position error (lower is better), NMPJPE=Normalized MPJPE (root alignment), PA-MPJPE = Procrustes alignment MPJPE, mean ± std), are presented in the table below.
> > > >
> > > > ### Hands Motion Estimation Error
> > > > | Metric | MPJPE | NMPJPE | PA-MPJPE |
> > > > | :--- | :--- | :--- | :--- |
> > > > | **4 cameras** | 94.12mm ± 337.52mm | 52.57mm ± 143.49mm | 17.16mm ± 29.33mm |
> > > > | **5 cameras** | 42.37mm ± 119.45mm | 33.3mm ± 75.18mm | 12.83mm ± 22.83mm |
> > > > | **All cameras**| 31.73mm ± 43.0mm | 29.91mm ± 42.6mm | 13.62mm ± 26.27mm |
> > > >
> > > > Increasing the number of cameras has a significant impact on the accuracy of the hand pose estimation. In particular, setups with only four global cameras have an error variance larger than the size of an average hand, making the setup highly unfitted for hand motion estimation. We will add this additional ablation in the revised manuscript.
> > > >
> > > > Please feel free to let us know if you have any additional follow-up questions, and we would be happy to address them further.

---

> > > > > ### Comment · Reviewer_tnRW · 2025-08-07
> > > > >
> > > > > Although I still don’t fully agree with the author’s responses, the author put a lot of effort into the discussion and added experiments to substantiate their claims, so I will change my score from “Borderline reject” to “Borderline accept.”

---

> > > > > > ### Author Response · Authors · 2025-08-07
> > > > > >
> > > > > > Thank you for acknowledging our work and providing thoughtful feedback. We truly appreciate your insights and constructive suggestions, which have helped us improve the paper.

---

### Official Review · Reviewer_oobx · 2025-07-03

**Rating:** 4
**Confidence:** 3

**Summary:**

The paper introduces X-Smart-Kitchen-30, a multimodal dataset designed to address critical gaps in human behavior understanding. Captured across synchronized ego- and exocentric cameras, the dataset features ego and exo videos, IMU data, eye gaze, and 3D hand/body poses from 29.7 hours of cooking activities by 16 participants. It includes 763 fine-grained action labels, yielding over 55,000 action segments, and supports four benchmarks: action recognition, segmentation, full-body motion generation, and a VQA task called Lemonade. By integrating visual, physical, and attentional modalities in natural, goal-directed scenarios, the dataset offers a rich, hierarchical representation of human behavior and aims to unify currently fragmented approaches in behavioral analysis.

**Dataset Code Accessibility:**

Yes

**Ethical Considerations:**

No, there are no or only very minor ethics concerns

**Final Justification:**

Thank you to the authors for their detailed and thoughtful responses. My concerns have been addressed. I intend to maintain my original score.

**Limitations Weaknesses:**

1. Regarding EgoExo4D, the paper reports its duration as 88.8 hours, which differs from the number in the original EgoExo4D paper. Is this based on a subset of the data? Additionally, EgoExo4D provides global positioning data (environment point cloud) and partial pose annotations for a portion of the dataset. It would be helpful to include a clearer comparison and discussion on these aspects.

2. A key strength of the proposed X-Smart-Kitchen-30 is its dense 3D body and hand pose annotations. However, the sample questions shown in the Lemonade benchmark seem relatively generic (e.g., action counting), similar to standard video QA tasks. It would be interesting to see how the 3D pose information could be leveraged to generate more pose- or motion-centric question-answer pairs that go beyond surface-level understanding.

3. In the full-body motion generation benchmark, the paper currently presents results from only two baseline models: T2M-GPT and MoMask. Including additional baselines or ablation studies would strengthen the evaluation and help demonstrate the versatility and challenge of the dataset.

**Strengths Contributions:**

1. The paper is well-written and easy to follow, with a clear structure and thorough explanation of the dataset and benchmarks.

2. The dataset provides annotations at both fine-grained and coarse-grained levels, which is valuable for advancing research in human behavior understanding.

3. The proposed benchmarks are extensive and multimodal, covering four key tasks: video-language question answering (VQA), action recognition, action segmentation, and full-body motion generation.

---

> ### Author Rebuttal · Authors · 2025-07-31
>
> We thank the reviewer for their thoughtful comments and the time and effort spent reviewing our paper. We appreciate that they recognized the potential of our dataset, particularly its behavioral annotations and multimodality.
>
> >1. “The paper reports its duration as 88.8 hours, which differs from the number in the original EgoExo4D paper.”
>
> Some of our metrics and statistics are extracted from Nymeria [1], which provides detailed video and motion annotations. The values for "Total hours," "Absolute positioning," and the presence of a "Parametric model" in our Table 1 are taken directly from Table 2 of the Nymeria paper (q/h, gp and pm).  Specifically, the "total activity hours" reported excludes the effect of the number of cameras and gives a fair comparison with other single-view datasets and motion capture datasets.
>
> >1. "… EgoExo4D provides global positioning data (environment point cloud) and partial pose annotations for a portion of the dataset.”
>
> We use the term "Absolute position" to refer to global positioning (derived from point clouds) that is consistent across different sessions. This will be clarified in the revised manuscript. All data in our work, spanning multiple sessions and participants, is globally aligned into a single 3D world coordinate system.
>
> Regarding hand and body poses, EgoExo4D only annotates keypoint-level and frame-level 3D poses. Our table reports the existence of a parametric model, the latter containing more information than keypoint representations. A continuous parametric model for motion representation is essential for motion understanding and modeling. We will add a more detailed comparison in our revised manuscript to make this distinction clearer.
>
> >2. “However, the sample questions shown in the Lemonade benchmark seem relatively generic (e.g., action counting), similar to standard video QA tasks.”
>
> The Lemonade benchmark is composed of three different categories. Behavior understanding, Long-term understanding, and Kinematic understanding. In the latter, we leverage the dense 3D body and hand pose to generate 9,463 kinematic-centered questions which are fundamentally different from other motion oriented MCQA benchmarks. Answers to these questions are typically numerical values extracted from kinematic features (speed, angles, distances) derived from the 3D pose data. Unlike many other MCQA datasets (MVBench [2], MLVU [3]) our answers are also derived from ground truth annotations (by humans) rather than LLMs. We are not aware of any “standard video QA tasks” that leverages multimodality and ground truth human manual annotations.
>
> >3. “In the full-body motion generation benchmark, the paper currently presents results from only two baseline models: T2M-GPT and MoMask.”
>
> We thank the reviewer for their remark. Firstly, we want to emphasize that we picked MoMask as it was SOTA on HumanML3D [4] and KIT [5] datasets. Meanwhile, T2M-GPT is another strong baseline that is widely used and can show the challenge of our benchmark.
>
> Secondly, we appreciate the reviewer’s comments on adding additional baselines. Due to time constraints, we add one more recent model named MARDM [6] that uses a continuous tokenizer to generate motions. We adapt the models in a similar way than for Momask and T2M-GPT to support full-body motion generation. Below are the results on our benchmark. We first re-trained the autoencoder tokenizer used by MARDM. We can see due to the removal of commitment loss, the reconstruction FID becomes lower compared to the VQ-VAE model.
>
> | Vocabulary  | Verbs  | Verbs     | Actions | Actions   |
> |-----------------|-----------|-------------|-----------|-------------|
> | Models        | rFID↓  | MPJPE↓ | rFID↓    | MPJPE↓ |
> |AutoEncoder| 0.347  |  0.293    | 0.404    | 0.295      |
>
> We then train MARDM-SiT (their best model) under both text and text-image conditions. We can see that MARDM-SiT overall performs slightly worse compared to Momask. This may be due to the higher dimensional properties of the whole-body representation, which makes it more challenging to learn the distribution of continuous representations. This experiment further supports the challenge of our dataset.
>
> | Condition  | Vocabulary | T1↑   | T2↑   | T3↑   | FID↓  | DIV↑  | MM↑   | MMd↓  |
> |---------------|-----------------|--------|--------|---------|---------|---------|----------|----------|
> | Text           | Verbs          | 0.262| 0.473| 0.637 | 2.242 | 7.590 |1.849 | 4.155  |
> | Text           | Actions       | 0.320| 0.548 | 0.650 | 1.230| 7.704 | 1.427 | 4.103 |
> | Text-Image| Verbs         | 0.187| 0.355| 0.466  | 1.498 | 6.579 | 1.581 | 4.691 |
> | Text-Image| Actions       | 0.255| 0.469| 0.597 | 0.917 | 7.176 | 1.322  | 4.725 |
>
> Upon acceptance, we will revise our paper and add the necessary experiments used in the rebuttal to the camera-ready paper. We hope that our responses have sufficiently addressed all comments.
>
> ## References
>
> [1] Ma, Lingni, et al. "Nymeria: A massive collection of multimodal egocentric daily motion in the wild." European Conference on Computer Vision. Cham: Springer Nature Switzerland, 2024.
>
> [2] Li, Kunchang, et al. "Mvbench: A comprehensive multi-modal video understanding benchmark." Proceedings of the IEEE/CVF Conference on Computer Vision and Pattern Recognition. 2024.
>
> [3] Zhou, Junjie, et al. "Mlvu: Benchmarking multi-task long video understanding." Proceedings of the Computer Vision and Pattern Recognition Conference. 2025.
>
> [4] Guo, Chuan, et al. "Generating diverse and natural 3d human motions from text." Proceedings of the IEEE/CVF conference on computer vision and pattern recognition. 2022.
>
> [5] Plappert, Matthias, Christian Mandery, and Tamim Asfour. "The kit motion-language dataset." Big data 4.4 (2016): 236-252.
>
> [6] Meng, Zichong, et al. "Rethinking Diffusion for Text-Driven Human Motion Generation: Redundant Representations, Evaluation, and Masked Autoregression." Proceedings of the Computer Vision and Pattern Recognition Conference. 2025.

---

> > ### Author Response · Authors · 2025-08-04
> > **Kind Reminder: Seeking Reviewer Feedback for Author/Reviewer Discussion Phase**
> >
> > Dear reviewer oobx,
> >
> > We hope this message finds you well. We would like to express our gratitude for the time and effort that you have dedicated to the review process of our submission.
> >
> > Furthermore, we are writing to kindly remind you that the rebuttal period is coming to a close. Your feedback and evaluation are crucial to the progress of our work, and we value your expertise and insights immensely.
> >
> > During rebuttal, we hope to have addressed all the queries and concerns you raised regarding our submission. Each point has been carefully considered, and we have provided detailed responses.
> >
> > We are eager to engage in further discussions with you on the responses and any other aspects of our study. Your feedback plays a pivotal role in shaping the direction of our research, and we are keen to hear your thoughts and suggestions.
> >
> > Thank you for your consideration.
> >
> > Best regards,
> >
> > Authors of submission #441

---

> > > ### Author Response · Authors · 2025-08-04
> > >
> > > We appreciate your questions regarding EgoExo4D, which further motivates to clarify our dataset's contributions:
> > >
> > > **Complementary Focus**: While EgoExo4D provides broad environmental coverage for skill understanding, X-Smart-Kitchen-30 addresses the orthogonal challenge of accurate motion capture for behavior modeling. These represent different research priorities within the broader human understanding ecosystem.
> > >
> > > **Technical Capabilities**: Our 10-camera setup enables continuous parametric pose models and dense hand tracking throughout 29.7 hours—capabilities that EgoExo4D's sparser camera configuration cannot achieve. This precision enables us to propose benchmarks for multimodal motion synthesis, and action segmentation. Our *text-to-motion generation benchmark* enables training models that can synthesize realistic cooking behaviors conditioned on natural language and visual context. It will also enable detailed future kinematic analysis
> > >
> > > **Study design**: Our controlled recipe repetition follows established experimental design principles (in behavioral science), enabling statistical analysis of behavioral variations across individuals and sessions. This controlled depth complements EgoExo4D's environmental breadth.
> > >
> > > **Bridge Vision-Motion Gap**: X-Smart-Kitchen-30 uniquely combines high-frequency multi-modal streams (100Hz IMU, 30fps parametric poses, multi-view RGBD) in a unified framework, addressing the current fragmentation between video and motion research communities. This enables researchers to leverage both our precise motion data and the environmental diversity of EPIC-Kitchens, EgoExo4D, and other datasets for comprehensive behavior understanding.

---

> > ### Comment · Reviewer_oobx · 2025-08-05
> >
> > Thank you to the authors for their detailed and thoughtful responses. All of my concerns have been addressed. I intend to maintain my original score.

---

> > > ### Author Response · Authors · 2025-08-07
> > >
> > > We're encouraged that you found all concerns addressed and appreciate your recognition of our 'technically solid' contributions. Your feedback has been invaluable in strengthening our work, particularly your suggestions about additional baselines, which we've now incorporated.
> > >
> > > Thank you!

---

### Decision · Program_Chairs · 2025-09-18

**Decision:**

Accept (poster)

**Comment:**

The paper presents X-Smart-Kitchen-30, a 29.7-hour multimodal dataset of cooking activities involving 16 participants performing four recipes. Data were collected using nine RGB-D cameras, a HoloLens 2 (ego video and eye gaze), and IMUs on tools/appliances, providing synchronized ego–exo video, gaze, inertial data, and dense 3D pose annotations. The dataset comprises 763 action classes and 55,361 labeled segments, averaging 33 segments per minute. The authors also introduce a new VQA benchmark, Lemonade, covering three categories: behavior understanding, long-term understanding/summarization, and motion/kinematics understanding.

The reviewers recognized that the eye gaze + full-body motion is particularly promising for new research directions. They also recognized that the paper is clearly written, well-structured, and visually engaging. Figures and statistics are detailed and informative.

The main concerns are its limited scale, restricted diversity, overlap with existing datasets, and under-exploitation of unique modalities.  Compared with EgoExo4D, for example, the data are collected in a single lab kitchen with only 4 recipes.  The IMU and gaze data seem to be useful for various tasks, but not fully explored in experiments.

In the author–reviewer discussion, the authors provided additional details and experimental results, addressing questions such as motion estimation errors with respect to the number of cameras and motion generation results using an additional baseline. The reviewers concluded that the paper merits acceptance. Accordingly, the AC also recommends acceptance.